# Comparative Evaluation of Physicochemical Properties, Microstructure, and Antioxidant Activity of Jujube Polysaccharides Subjected to Hot Air, Infrared, Radio Frequency, and Freeze Drying

**Bengang Wu [1,2]**, **Chengcheng Qiu [1]**, **Yiting Guo [1,2,\*]**, **Chunhong Zhang [3]**, **Dan Li [3]**, **Kun Gao [1]**, **Yuanjin Ma [1]** and **Haile Ma [1,2]**

1   School of Food and Biological Engineering, Jiangsu University, 301 Xuefu Road, Zhenjiang 212013, China
2   Institute of Food Physical Processing, Jiangsu University, 301 Xuefu Road, Zhenjiang 212013, China
3   Naval Medical Center of PLA, Naval Medical University (Second Military Medical University), Shanghai 200433, China
\*   Correspondence: 1000005604@ujs.edu.cn

**Abstract:** In this study, we used four drying methods (hot air, freezing, infrared, and radio frequency) to dry fresh jujube and its polysaccharide extracts by a two-step drying method, and the effects of the drying methods on the physical and chemical properties, structural properties, and antioxidant activity of jujube polysaccharides were studied. The results showed significant differences in the yield, drying time, monosaccharide content, molecular weight, apparent viscosity, thermal stability, and microstructure of the polysaccharides treated under the different drying methods. In contrast, no significant differences in the monosaccharide composition and functional groups of the polysaccharide samples obtained from the different drying methods were observed. Among all the tested methods, the freeze-drying extraction rate was the highest, reaching $4.52 \pm 0.19\%$, while its drying time was the longest. Although the extraction rate of radio frequency drying was only $3.55 \pm 0.21\%$, the drying time was the shortest, compared with hot air drying, the drying time was reduced by 76.67–83.29%, and the obtained polysaccharides exerted good antioxidant activity. Therefore, radio frequency drying is a potential polysaccharide extraction and drying technique, and this study can provide a theoretical basis for its industrial production.

**Keywords:** jujube polysaccharide; drying methods; physicochemical property; structural characteristic; antioxidant activity

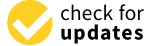

## 1. Introduction

Jujube is a fruit of the plant jujube tree, a plant of the family rhamnaceae. It is a specialty species native to China and has a long history [1]. Jujube has high nutritional and medicinal value, such as nourishing the yin and kidney, strengthening the body, softening blood vessels, and enhancing the physiological function of each viscus [2]. Jujube extracts have been proven to have many bioactive substances, among which polysaccharide is considered one of the most important bioactive components [3]. Modern medical research shows that jujube polysaccharides (JPSs) have antioxidant, antiviral, antitumor, and other functions, so the development prospect of jujube polysaccharides is extensive [4].

Fresh jujube has high water content, so it easily deteriorates, leading to the loss of its nutrients, which is not conducive to further extraction of bioactive components [5]. Traditional drying techniques, such as hot air drying (HAD) and freeze drying (FD), are easy to control. However, these traditional drying techniques have many problems, such as high energy consumption, low drying efficiency, a long drying time, low quality of the products after drying, and expensive freeze-drying equipment. Therefore, in this case, it

is imperative for us to find a new drying technology with the attributes of a short drying time, high drying efficiency, and good product quality. Radio frequency drying (RFD) is a new drying technique that generates heat inside food through ion conduction and dipole rotation, so that water can be evaporated [6]. Infrared drying (IRD) is a new method that radiates materials with the energy generated by an infrared emitter, and the infrared waves are absorbed and converted into heat energy by the materials [7]. These new drying methods have the advantages of a short drying time, high efficiency, and good product quality, which can be a complete or partial alternative to the traditional methods. A large number of studies have revealed that different drying techniques have significant effects on the structural characteristics and biological activities of polysaccharides derived from loquat leaves [8], dandelion [9], and mulberry leaves [10]. However, the effects of these different drying techniques on the structural characteristics and biological activities of the polysaccharides extracted from jujube are still unclear.

In this study, four drying methods (HAD, FD, IRD, and RFD) were used to perform a two-step drying treatment on fresh jujube to extract its polysaccharides (Figure 1). The effects of these drying methods on the physicochemical and structural properties of jujube polysaccharides in terms of thermogravimetric characteristics, rheological properties, and FT-IR spectroscopy were determined, and its antioxidant activity (DPPH and ABTS) was also studied. The detailed results can contribute to a better knowledge of jujube polysaccharide extraction and further provide more information for the practical industrial application of this two-step drying method.

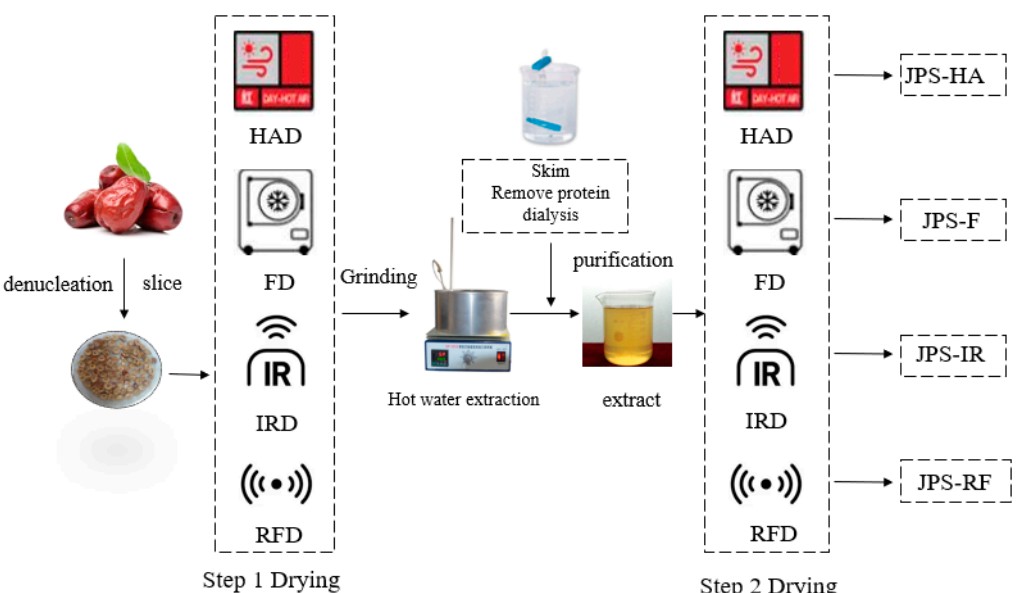

**Figure 1.** Schematic diagram of the two-step drying method of jujube polysaccharide extraction.

## 2. Materials and Methods

### 2.1. Materials and Chemicals

Fresh jujubes (*Zizyphus jujuba* Mill., Huizao) were collected from Xinjiang, China. Monosaccharides and uronic acids were purchased from Aladdin Biochemical Technology Co., Ltd. (Shanghai, China). Other chemicals used in this study were all analytical grade and purchased from Sinopharm (Shanghai, China).

### 2.2. Drying Procedures

Firstly, the jujubes were denucleated, sliced, and divided into four groups according to different drying processes, which were hot air drying (HAD), freeze drying (FD), infrared drying (IRD), and radio frequency drying (RFD). Briefly, in the FD process, jujube slices were placed in a glass Petri dish (5 mm diameter) and were freeze-dried for 48 h at $-40\ ^\circ$C.

In the HD process, jujube slices were placed into an oven at 70 °C. In the IRD process, the infrared device power was adjusted to 900 W, and the corresponding surface temperature of the jujube slices reached 70 °C. In the RFD process, well-calibrated radio frequency equipment (HGJL-6RFS, Hefei Hagong Jinlang Equipment Technology Co., Ltd., Hefei, China) was used to dry jujube slices at an anode voltage of 70%, and the distance of the sample tray to the heater was set to 6 cm. The endpoints of the four drying methods were chosen as when the jujube samples reached a constant weight. The used time and final moisture content of the samples for each method were recorded.

### 2.3. Extraction of Polysaccharides from Jujube

Hot water extraction of polysaccharides from jujube was carried out by the previously reported method, with slight modification [2]. Briefly, the dried jujube was ground and sifted through 60 mesh. An appropriate amount (50 g) of the sample was taken and treated with petroleum ether and 80% (*v*/*v*) ethanol for 2 h, respectively, to remove oil and small molecules. Then, the obtained sample was dried and extracted with 500 mL of deionized water at 90 °C for 4 h. The supernatant was collected after centrifuging and precipitated with four volumes of anhydrous ethanol at 4 °C overnight. Subsequently, the collected precipitates were redissolved in appropriate amount of deionized water and dialyzed with deionized water (molar mass cutoff: 3.5 kDa) for 4 h. Finally, the dialysate was subjected to HAD, FD, IRD, and RFD, to obtain the final polysaccharide powder, and the obtained products were labeled as JPS-H, JPS-F, JPS-IR, and JPS-RF.

### 2.4. Determination of Physicochemical Properties of Polysaccharides

#### 2.4.1. Analysis of Chemical Components

The total sugar content of polysaccharides was determined by the phenol-sulfuric acid method, with D-glucose as a standard [11]. The protein content was identified by the Coomassie brilliant blue G-250 method. The carbazole sulfuric acid method was used to determine the uronic acid content of polysaccharides [12]. The polyphenol content in polysaccharides was determined by the Folin–Ciocalteu method [13]. All tests were performed in triplicate.

#### 2.4.2. Monosaccharide Composition and Content of JPSs

The monosaccharide composition was determined by high-performance liquid chromatography (HPLC) (LC-20AT, Shimadzu Co., Ltd., Kyoto, Japan) with PMP precolumn derivatization [14]. Briefly, 2 mg of sample was accurately weighed into a 10 mL hydrolytic tube, and 2 mL of 2 M trifluoroacetic acid (TFA) was added, sealed with nitrogen, and hydrolyzed at 110 °C for 6 h. After hydrolysis, rotation evaporation was used to dry the obtained solution to remove TFA. After that, 3 mL of methanol was added to the tubes, and the step of evaporation–methanol washing was repeated 4 times to ensure that all TFA was removed. The hydrolysate was dissolved in 2 mL of distilled water, and 450 μL of 0.5 M PMP solution and 450 μL of 0.3 M NaOH were added into the hydrolysate sequentially. After mixing well, the mixture was incubated at 70 °C for 40 min, and then 450 μL of 0.3 M HCl was added to terminate the reaction. The hydrolyzed sample was filtered through a 0.45 μm aqueous membrane and analyzed by HPLC with an XDB-C18 column (4.6 × 50 mm, 25 cm). A mixture of maltose, rhamnose, arabinose, galactose, glucose, mannose, xylose, and glucuronic acid, at a series of concentrations, was selected as external standards for sugar identification and quantitation. Three replications were conducted for each test.

#### 2.4.3. Molecular Weight of JPSs

The weight-average molecular weight (Mw), number-average molecular weight (Mn), molecular weight distribution (Mw/Mn), and root-mean-square rotation radius (Rg) of JPSs were determined by size-exclusion chromatography (HPSEC) equipped (G136A, Agilent Technologies, Santa Clara, CA, USA) with a MultiAngle Laser Light scattering

(SEC-MALLS) detector and a differential refraction detector (RI) detector, according to the method of Gu et al. [15]. An accurate amount of sample (4 mg) was weighed and dissolved in 4 mL of 0.1 M NaCl solution containing 0.02% $NaN_3$ (*w/v*) (mobile phase). The solution was filtered through a 0.45 μm aqueous membrane and injected into the HPSEC for detection. Data were analyzed using Astra 6.1.7 software.

### 2.4.4. Thermogravimetric Analysis (TGA)

The thermogravimetric analysis (TGA) determination of JPSs followed the method of Liu et al. [16]. The TGA was conducted on a DTG-60 thermal analyzer (Shimadzu, Kyoto, Japan) to investigate the thermal properties of polysaccharides. Briefly, the samples (10 mg) were placed in a sealed aluminum pan and heated from 30 to 700 °C at a heating rate of 10 °C/$min^{-1}$ under nitrogen. The empty, sealed aluminum pan was utilized as a control.

### 2.4.5. Determination of Rheological Properties

The rheological properties of JPSs were determined at room temperature (25 °C) using a DHR-1 rheometer (Discovery HR-1, TA Corporation, USA) equipped with a plate (40 mm diameter, 1 mm gap). Determination of apparent viscosity (η) of samples was performed in the shear rate range of 0.1–100 $s^{-1}$ [17].

### 2.4.6. FT-IR Spectroscopy Analysis

The dried JPS samples were analyzed by an ATR-FTIR spectrometer (Semefi Technology Co., Ltd., Huludao, China). For all samples, 32 scans were performed at 4 $cm^{-1}$ resolution in a scanning range of 4000–600 $cm^{-1}$. The obtained spectrograms were analyzed by OMNIC (Thermo Inc., Waltham, MA, USA).

### *2.5. Scanning Electron Microscopy (SEM)*

Scanning electron micrographs of JPSs were conducted according to the modified method of Wang et al. [18]. The dried polysaccharide powders were mounted onto a specimen stub, with the assistance of conductive tape, and sputter-coated with gold. The well-prepared samples were photographed using a Hitachi S-3400N scanning electron microscope (Hitachi Inc., Tokyo, Japan), and the images were captured at a voltage of 15 kV.

### *2.6. Antioxidant Activities of JPS*

### 2.6.1. Determination of DPPH Radical-Scavenging Assay

The DPPH radical-scavenging activity of dried JPS powders was determined based on the method of Wang et al. [19]. A series of concentrations of JPS solution (1 mL) was mixed with 3 mL of 0.2 mM DPPH ethanol solution. After shaking well, the mixture was incubated in dark at 37 °C for 30 min. The absorbance was measured at 517 nm with the absolute ethanol as the control. The DPPH radical-scavenging activity was calculated as follows:

$$DPPH\ radical\text{-}scavenging\ activity\ (\%) = \left(1 - \frac{A_1}{A_0}\right) \times 100\% \tag{1}$$

where $A_1$ is the absorbance of JPS, and $A_0$ is the absorbance of the control.

### 2.6.2. Determination of ABTS Radical-Scavenging Activity

The ABTS radical-scavenging activity was basically measured as described by Chen et al. [20]. A stock solution of ABTS (7 mM) was prepared by dissolving ABTS in ethanol. Then, potassium persulfate was added to the prepared stock solution (final concentration was 2.45 mM), and the mixed solution was allowed to stand overnight at ambient temperature in darkness. The ABTS working solution was prepared by diluting the stock solution until the absorbance measured between 0.7 to 0.734. Subsequently, an aliquot of 2 mL of JPS solution was mixed with ABTS working solution in equal volume at room temperature

in darkness for 10 min. Then, the absorbance was recorded at 734 nm, with ABTS working solution as the control. The ABTS radical-scavenging activity was calculated as follows:

$$ABTS\ radical\text{-}scavenging\ activity\ (\%) = \frac{A_0 - A_1}{A_0} \times 100\% \qquad (2)$$

where $A_1$ is the absorbance of JPS, and $A_0$ is the absorbance of the control.

### 2.7. Statistical Analysis

All experiments were performed in three replicates, and the experimental results were expressed as the mean ± standard deviation (SD). Statistically significant differences in results were determined by one-way analysis of variance (ANOVA) with Duncan's multiple range test. SPSS 26.0 (IBM Corporation, Chicago, IL, USA) was used for all statistical analyses, at the significance level of $p = 0.05$.

## 3. Results and Discussion

### 3.1. Yields and Chemical Compositions of JPSs

As shown in Table 1, the extraction rate of JPS-F was the highest among all the tested methods, reaching 4.52%. Compared with HAD, the extraction rate of FD was increased by 53.2%, which was due to the fact that FD rendered porous structures into the jujube tissues; as a result, it was easier for water to penetrate the jujube tissues during the extraction process, though this method required the longest duration compared with the other methods [21]. Moreover, the lower processing temperature and less oxygen content in the environment were more conducive to extracting polysaccharides during the FD process [9]. As for extraction time, the longest duration of FD was used to dry fresh jujube to a constant weight, reaching 48 h. In contrast, the durations of IRD and RFD required to dry jujube to the same water content as FD decreased significantly ($p < 0.05$).

**Table 1.** Extraction rate, drying duration, and chemical composition of JPSs treated with different drying methods.

| Sample | JPS-HA | JPS-F | JPS-IR | JPS-RF |
|---|---|---|---|---|
| Yield (%) | 2.95 ± 0.08 a | 4.52 ± 0.19 c | 3.10 ± 0.19 a | 3.55 ± 0.21 b |
| Step 1 drying time (h) | 4.25 ± 0.35 b | 48.00 ± 0.00 c | 0.88 ± 0.06 a | 0.71 ± 0.06 a |
| Step 2 drying time (h) | 5.10 ± 0.14 b | 48.00 ± 0.00 c | 1.42 ± 0.12 a | 1.19 ± 0.02 a |
| Total sugar content (%) | 66.19 ± 0.81 a | 76.06 ± 1.24 d | 69.33 ± 0.93 b | 74.26 ± 0.98 c |
| Total uronic acid content (%) | 11.95 ± 0.004 a | 15.65 ± 0.005 b | 18.78 ± 0.002 c | 19.18 ± 0.003 c |
| Total phenol content (%) | 0.55 ± 0.07 a | 1.17 ± 0.10 b | 0.61 ± 0.07 a | 1.04 ± 0.13 b |
| Protein content (%) | 2.93 ± 0.13 b | 1.41 ± 0.13 a | 2.83 ± 0.54 a | 1.69 ± 0.27 c |

Note: Values are means ± standard deviation. Values followed by different letters in each column indicate significant differences ($p < 0.05$).

In general, the extracted crude polysaccharides contained many substances, such as proteins and polyphenols, which might combine with other components and show various activities. Therefore, it is necessary to measure the chemical composition of crude polysaccharides. As shown in Table 1, the total sugar contents were 66.19%–76.06%, indicating that polysaccharide was the dominant constituent in the extract. In addition, glucuronic acid was found in the extracts dried with these four methods, indicating the presence of pectin-like polysaccharides in jujube. Similar results were obtained in polysaccharides extracted from loquat and okra [22]. Although the samples were deproteinized and dialyzed, they still contained a small amount of protein and polyphenols, which may be due to the presence of protein–polysaccharide and polyphenol–polysaccharide complexes in the extracted polysaccharides. Such complexes can enhance the biological activity of polysaccharides to some extent [23].

### 3.2. Monosaccharide Compositions and Molecular Weights of JPSs

Monosaccharide is the natural basic unit that can determine the structures and characteristics of polysaccharides [24]. As shown in Table 2, the standard monosaccharides used were maltose (Mal), rhamnose (Rha), arabinose (Ara), galactose (Gal), glucose (Glu), glucuronic acid (GluA), mannose (Man), and xylose (Xyl). In general, the different drying methods had no effect on the monosaccharide composition of JPS. However, they had a significant effect on the monosaccharide content, which is consistent with the study of Chen et al. (2020), who investigated the effect of different drying methods on the structural characteristics of polysaccharides from bamboo shoots [25]. From Table 2, it can be seen that the seven tested monosaccharides and a uronic acid all can be detected in the polysaccharides that were extracted using the different drying methods. Among these monosaccharides, Mal was the predominant sugar, followed by Ara. The difference in monosaccharide content may be related to the oxygen content and environmental temperature. Compared with HAD, the Glu and Ara contents in the JPSs treated with FD, IRD, and RFD were significantly ($p < 0.05$) reduced, which is probably due to the hydroxyl oxidation and intermolecular hydrogen bond breakage of the polysaccharides in the environment, affecting the monosaccharide contents [26].

**Table 2.** Monosaccharide constituents and molecular weight of JPSs treated with different drying methods.

| Sample | JPS-HA | JPS-F | JPS-IR | JPS-RF |
|---|---|---|---|---|
| Mw ($\times 10^5$ Da) | 1.62 ($\pm$1.38%) | 0.87 ($\pm$1.11%) | 1.26 ($\pm$1.33%) | 1.20 ($\pm$1.87%) |
| Mn ($\times 10^5$ Da) | 1.20 ($\pm$6.09%); | 0.77 ($\pm$4.93%) | 0.93 ($\pm$6.02%) | 0.92 ($\pm$1.85%) |
| Mw/Mn | 1.34 ($\pm$6.24%) | 1.13 ($\pm$5.05%) | 1.36 ($\pm$6.17%) | 1.30 ($\pm$2.63%) |
| Rg (nm) | 38.7 ($\pm$0.1%) | 27.9 ($\pm$0.2%) | 38.0 ($\pm$0.1%) | 30.1 ($\pm$0.1%) |
| Mal | 40.81 $\pm$ 1.54 b | 36.18 $\pm$ 0.00 ab | 39.41 $\pm$ 0.65 ab | 34.73 $\pm$ 3.51 a |
| Rha | 3.89 $\pm$ 0.00 b | 3.74 $\pm$ 0.06 b | 3.39 $\pm$ 0.18 a | 3.44 $\pm$ 0.06 a |
| Gal | 6.81 $\pm$ 0.02 b | 6.90 $\pm$ 0.02 b | 5.87 $\pm$ 0.32 a | 6.26 $\pm$ 0.03 a |
| Ara | 10.33 $\pm$ 0.15 c | 9.18 $\pm$ 0.36 b | 8.35 $\pm$ 0.35 a | 8.36 $\pm$ 0.11 a |
| Glu | 5.03 $\pm$ 0.05 c | 3.37 $\pm$ 0.03 a | 4.09 $\pm$ 0.20 b | 3.15 $\pm$ 0.06 a |
| GluA | 3.52 $\pm$ 0.03 a | 4.36 $\pm$ 0.35 a | 2.31 $\pm$ 1.55 a | 3.11 $\pm$ 1.57 a |
| Man | 1.69 $\pm$ 0.00 b | 1.75 $\pm$ 0.00 c | 1.55 $\pm$ 0.04 a | 1.70 $\pm$ 0.00 bc |
| Xyl | 1.91 $\pm$ 0.04 a | 1.95 $\pm$ 0.09 a | 1.82 $\pm$ 0.02 a | 1.90 $\pm$ 0.04 a |

Note: Values are means $\pm$ standard deviation. Values followed by different letters in each column indicate significant differences ($p < 0.05$).

JPSs' molecular weight plays an important role in their functional properties and biological activity [24]. The molecular weight of polysaccharides obtained from the different drying methods was analyzed, and the data are shown in Table 2. The larger the Mw/Mn value of the polydispersion coefficient is, the wider the range of molecular weight distribution, which indicates that the polymer is in a mixed state and its purity is relatively low. The Mw/Mn value of JPS-F was the smallest, indicating that the range of its molecular weight distribution is the smallest among these four polysaccharides. The Mw values of JPS-HA, JPS-F, JPS-IR, and JPS-RF are $1.62 \times 10^5$, $0.87 \times 10^5$, $1.26 \times 10^5$, and $1.20 \times 10^5$ Da, respectively. Additionally, the smallest molecular weight of JPS-F also indicated that FD-extracted polysaccharides are loosely bound to other macromolecules within or between cells [27]. This is probably because the glycosidic bonds were broken under low temperature and low oxygen conditions, which further leads to the reduction of Mw.

### 3.3. Thermal Properties of JPSs

Figure 2A shows the thermal property of JPSs obtained under different drying conditions. The thermogravimetric analysis (TGA) curves of all JPSs were similar in shape over a defined temperature range (30 to 700 °C). In the first stage (below 250 °C), the free and bound water of the JPSs evaporated, and the weight loss was about 20%. The second

stage of thermal degradation occurred at about 250~500 °C, which is mainly because of the decomposition of polysaccharide chains and the rupture of hydrogen bonds as well as the degradation of thermal unstable functional groups, resulting in a fast weight loss (about 40%). In the final stage, when the temperature exceeds 500 °C, the weight loss of the JPSs remains constant [16]. According to Figure 2A, JPS-F has the worst thermal stability among all the polysaccharides, which may be related to its flocculent porous structure.

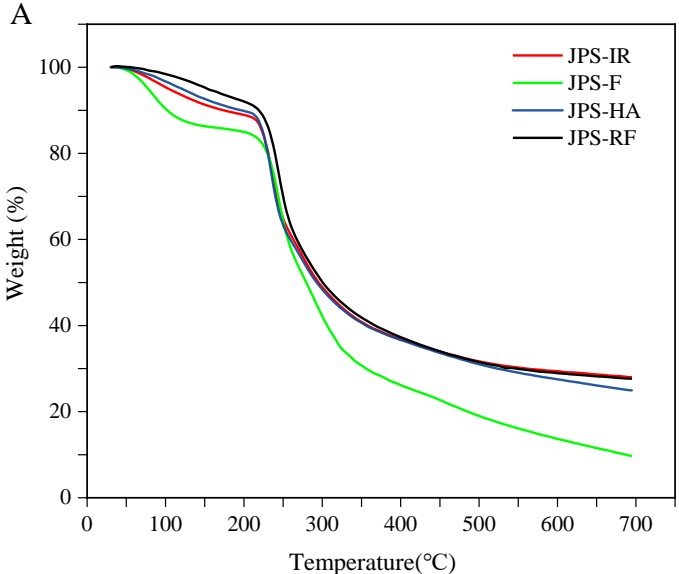

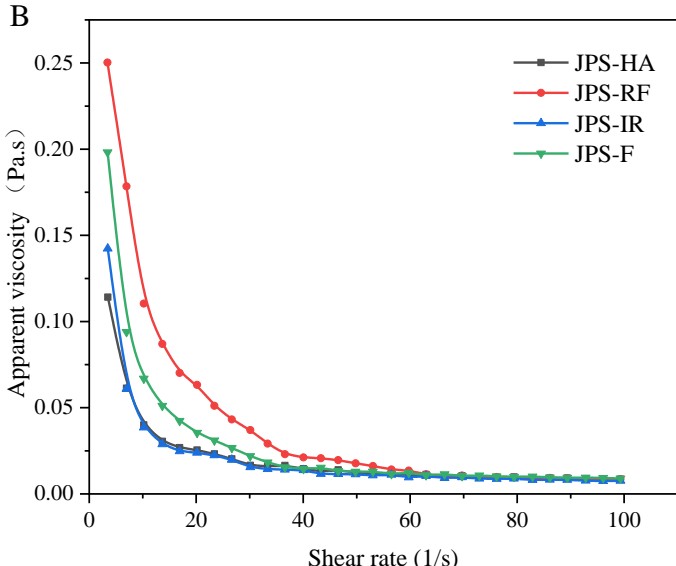

**Figure 2.** *Cont.*

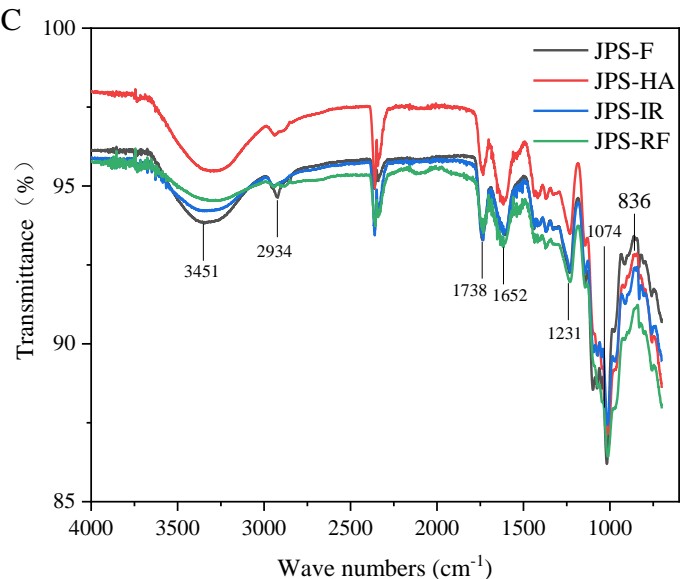

**Figure 2.** Thermogravimetric analysis (**A**), apparent viscosity (**B**), and FT-IR spectra (**C**) of JPSs dried under different methods.

*3.4. Rheological Properties of JPSs*

Viscosity is a vital property of polysaccharides that can influence their bioactivities. The flow behavior of the JPSs is displayed in Figure 2B, which shows that the apparent viscosity decreased with the shear rate increasing from 0.1 to 100 s$^{-1}$. Non-Newtonian shear-thinning behaviors could be found in JPS solutions at a low shear rate range (0.1–50 s$^{-1}$), while nearly Newtonian flow behavior was found at a high shear rate range (50–100 s$^{-1}$). The flow behavior alterations result from many reasons, and a large number of experimental studies have shown that the shear-thinning behavior of polysaccharides was related to the untangling of molecular chains in solution [28,29]. The results showed that the apparent viscosity of the JPSs was affected by the different drying methods. Among them, JPS-RF has the highest apparent viscosity, which is consistent with the results of Li et al. [9], who studied the effects of RF drying on dandelion polysaccharide. This result might be related to the molecular weight, monosaccharide composition, and uronic acid content of JPS-RF.

*3.5. FT-IR Spectra of JPSs*

FT-IR spectra can be used to efficiently and rapidly characterize the structural groups of polysaccharides. As presented in Figure 2C, the FT-IR spectra of JPS-HA, JPS-F, JPS-IR, and JPS-RF were similar, which indicates that the functional groups of these four JPSs had not changed during the different drying processes. It can be seen that these crude JPSs all exhibited a broadly stretched intense peak at 3451 cm$^{-1}$, for the hydroxyl stretching vibration, and a weak C-H stretching vibration at 2934 cm$^{-1}$. The absorption peak at 1738 cm$^{-1}$ was the stretching vibration of the esterified carboxylic groups [30]. Moreover, the strong absorption peak at 1652 cm$^{-1}$ was the C-O, demonstrating that the JPSs were acidic polysaccharides. Absorptions between 1000 and 1100 cm$^{-1}$ showed the stretching vibrations of the pyranose ring. Moreover, the peak at 836 cm$^{-1}$ in the JPSs indicated the existence of $\alpha$-type glycosidic linkage [10].

*3.6. Surface Morphology of JPSs*

As a visual analytical technique, SEM can be directly used to observe the microscopic characteristics of polysaccharides [24]. SEM images of JPS-HA, JPS-F, JPS-IR, and JPS-RF are shown in Figure 3. The surface structures of the polysaccharides extracted by the different drying methods differed. Among them, JPS-HA, JPS-IR, and JPS-RF showed a flake structure, and JPS-HA also exhibited many pores, which may be caused by long-term

exposure to a high-temperature environment. As the temperature rose, the water content in the jujube evaporated gradually, and the internal structure of the tissue was dislocated and irreversibly damaged. In the FD process, water was released by sublimation, which renders JPS-F a loose and porous structure.

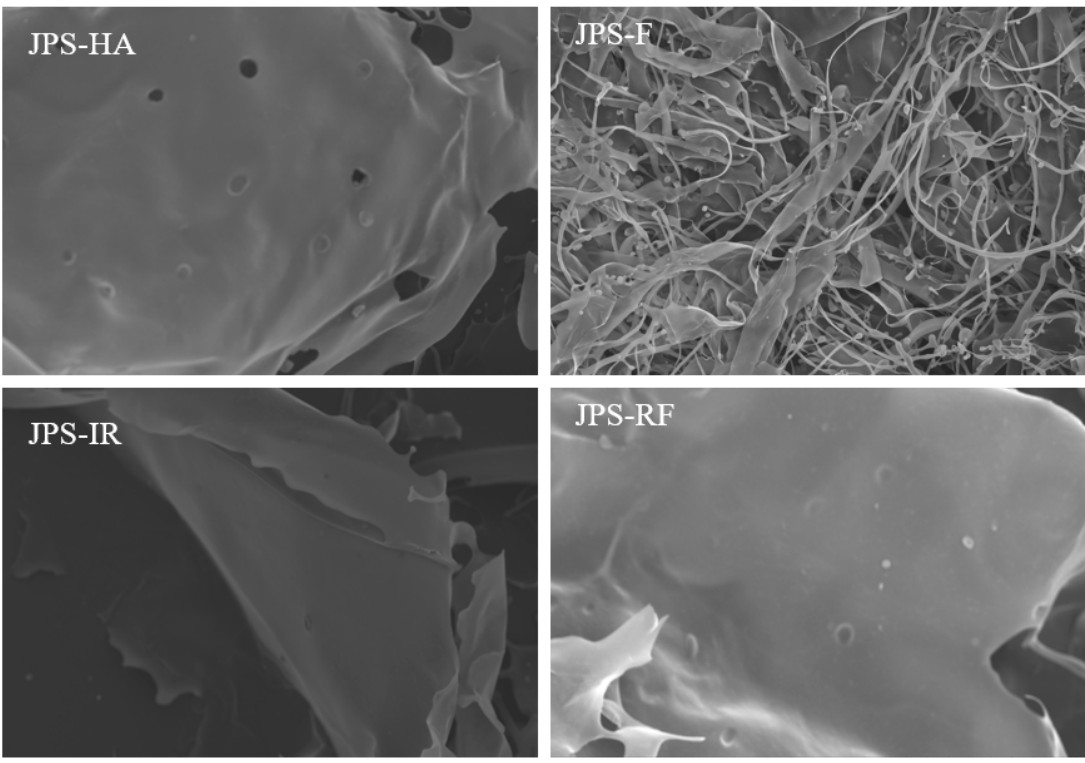

**Figure 3.** Scanning electron micrographs (500×) of four polysaccharides obtained from different drying methods.

### 3.7. Antioxidant Activity of JPSs

DPPH and ABTS radical scavenging methods were used to evaluate the antioxidant capacity of the JPSs extracted by the different drying methods [31]. As shown in Figure 4A, polysaccharides obtained from the four drying methods all showed good DPPH free radical-scavenging activity at different concentrations (0–1.0 mg/mL), and the antioxidant activity of JPSs increased with the increase in polysaccharide concentration during the entire process. When the concentration reaching 1 mg/mL, the four JPSs all showed their highest antioxidant activity, which was 59.42% (JPS-HA), 63.07% (JPS-F), 67.99% (JPS-IR), and 70.60% (JPS-RF). Since the DPPH scavenging activity is proportional to the JPS concentration, the $IC_{50}$ value can be selected to compare the antioxidant activity of the polysaccharides treated with the different drying methods [19]. As can be seen in Figure 4B, among the four JPSs, the $IC_{50}$ value of JPS-RF was the lowest, while that of the JPS-HA sample was the highest, indicating that JPS-RF could scavenge 50% of the free DPPH radicals at a lower concentration. Figure 4C shows the ABTS free radical scavenging capacity of JPS-HA, JPS-F, JPS-IR, and JPS-RF. The scavenging capacity of JPSs for ABTS radicals exhibited a similar tendency to the DPPH radical scavenging ability, which increased with the increase in the JPS concentration. The $IC_{50}$ values are shown in Figure 4D. The $IC_{50}$ value of the traditional drying method (HAD)-obtained JPSs showed no significant difference from the FD method, while the $IC_{50}$ of JPS-RF was significantly lower than that of the other methods' treated samples. Similar to the DPPH scavenging ability, the lowest value of $IC_{50}$ for the ABTS test was observed in the RF-dried jujube polysaccharides, which was 0.21 mg/mL.

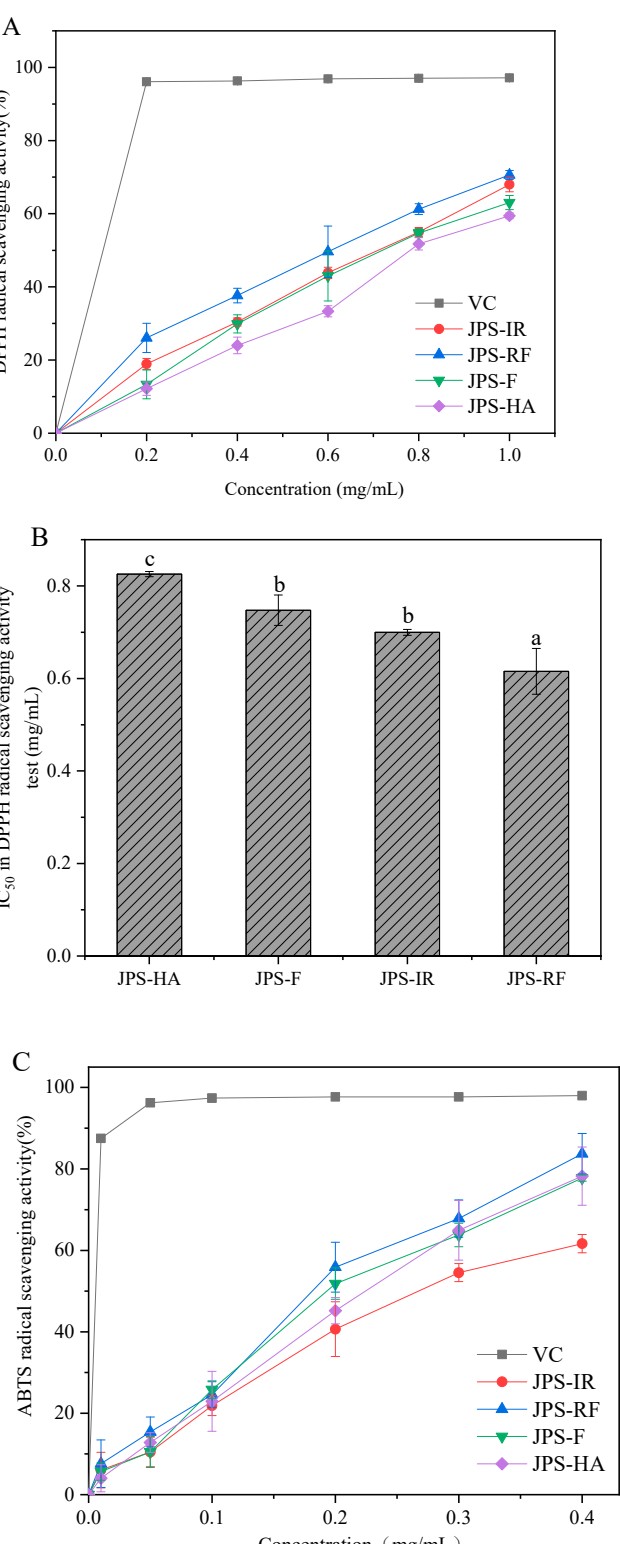

**Figure 4.** *Cont.*

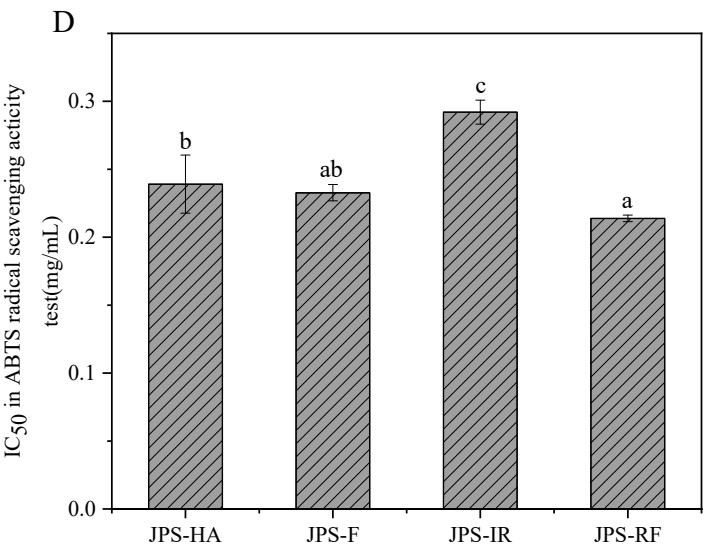

**Figure 4.** Effects of different drying methods on the antioxidant ability of JPSs. DPPH radical-scavenging activity (**A**) and its $IC_{50}$ value (**B**); ABTS radical-scavenging ability (**C**) and its $IC_{50}$ value (**D**). Bars with different letters differ significantly ($p < 0.05$).

By combining the results of the DPPH and ABTS tests, it was found that RFD technology is a promising drying method for extracting polysaccharides from jujube with high antioxidant activity, among the tested four methods. This result is consistent with the study of [9], who treated dandelion with different drying methods to study the structural activity of its polysaccharides. There are many factors affecting the antioxidant activity of jujube polysaccharides, such as the composition and proportion of monosaccharides, molecular weight, polyphenol content, and uronic acid content [10,32].

## 4. Conclusions

The effects of different drying techniques on the physical and chemical properties, functional structure, and antioxidant activity of JPSs were studied and compared. The results showed that different drying techniques not only affected the drying time of jujube and jujube polysaccharides but also significantly affected the physicochemical properties and antioxidant activity of jujube polysaccharides. The extraction time, molecular weight, monosaccharide composition, apparent viscosity, uronic acid content, and polyphenol content of the JPSs obtained from the different drying techniques were different. Compared with HAD, the polysaccharides obtained by RFD improved the drying efficiency and showed strong antioxidant activity. The current results can provide a theoretical basis for extracting and drying polysaccharides from jujube using this two-step drying method in industrial application.

**Author Contributions:** Conceptualization, B.W. and Y.G.; methodology, C.Q.; software, K.G.; formal analysis, Y.M.; data curation, C.Z.; writing—original draft preparation, C.Q. and B.W.; writing—review and editing, Y.G.; visualization, D.L.; supervision, Y.G.; project administration, H.M. All authors have read and agreed to the published version of the manuscript.

**Funding:** China Postdoctoral Science Foundation (Nos. 2021M700908 and 2022TQ0128) and the Key Laboratory of Storage of Agricultural Products, Ministry of Agriculture and Rural Affairs (No. kf2022002).

**Institutional Review Board Statement:** Not applicable.

**Informed Consent Statement:** Not applicable.

**Data Availability Statement:** Not applicable.

**Conflicts of Interest:** The authors declare that they have no conflict of interest.

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
