# Peer review of "Comparative Evaluation of Physicochemical Properties, Microstructure, and Antioxidant Activity of Jujube Polysaccharides Subjected to Hot Air, Infrared, Radio Frequency, and Freeze Drying"

_agriculture, doi:10.3390/agriculture12101606_

Round 1
Reviewer 1 Report
Reviewer comment: 1. The full text chart format should be unified, and the format of saliency annotation in the table should be consistent. like Table 1 and Table 2.
Reviewer comment: 2. There are many errors in the citation of the reference in the text, the authors should check the information carefully before submission. like line 164.
Reviewer comment: 3. In addition, authors should provide the full name of the abbreviations when it appeared first time in abstract or text. Please check it throughout the manuscript.
Reviewer comment: 4. In line 309, Remove any extra Spaces in the sentence and check the full text.
Author Response
Reviewer comment: 1. The full text chart format should be unified, and the format of saliency annotation in the table should be consistent. like Table 1 and Table 2.
Our response: We have unified the chart format throughout the text.
Reviewer comment: 2. There are many errors in the citation of the reference in the text, the authors should check the information carefully before submission. like line 164.
Our response: Thank you very much for your kind reminder. We have checked and revised the whole manuscript.
Reviewer comment: 3. In addition, authors should provide the full name of the abbreviations when it appeared first time in abstract or text. Please check it throughout the manuscript.
Our response: We have reviewed the full text
Reviewer comment: 4. In line 309, Remove any extra Spaces in the sentence and check the full text
Our response: We have reviewed the full text and removed excess white space.
Reviewer 2 Report
The manuscript is well written. The whole article has a reasonable structure, strong logic and contains some unique insights. After the article has been edited, I suggest that the manuscript be published in the journal.
Here are some suggestions for this paper.
>In line 92, you used petroleum ether and ethanol to remove oil and small molecules. Do they affect the edibility of the polysaccharide?
>In line 136, is there a problem with the units of heating rate?
>The formatting of figures and tables should be more rigorous. Notice the table on line 238.
>In line 316, please explain "This result is consistent with the study of ".
>In line 183, Add a space between words.
Author Response
The manuscript is well written. The whole article has a reasonable structure, strong logic and contains some unique insights. After the article has been edited, I suggest that the manuscript be published in the journal.
Here are some suggestions for this paper.
Reviewer comment: 1. In line 92, you used petroleum ether and ethanol to remove oil and small molecules. Do they affect the edibility of the polysaccharide?
Our response: Thank you for your advice. Firstly, it will not affect the edible ability of polysaccharide. Petroleum ether and ethanol are common processing AIDS in food. Secondly, for the removal of small molecules, we mainly use dialysis bag dialysis.
Reviewer comment: 2. In line 136, is there a problem with the units of heating rate?
Our response: We have checked and revised the whole manuscript.
Reviewer comment: 3. The formatting of figures and tables should be more rigorous. Notice the table on line 238.
Our response: We have unified the chart format throughout the text.
Reviewer comment: 4. In line 316, please explain "This result is consistent with the study of ".
Our response: We measured the antioxidant activity of polysaccharides by DPPH and ABTS, and found that RF drying technology could improve the antioxidant activity of polysaccharides, which was consistent with the study of Li et al.
Reviewer comment: 5. In line 183, Add a space between words.
Our response: We have revised it.
Reviewer 3 Report
I have gone through the MS entitled "Comparative evaluation of physicochemical properties, micro-structure, and antioxidant activity of jujube polysaccharides subjected to hot air, infrared, radiofrequency and freeze drying". The authors used four drying methods (hot air, freezing, infrared, and radiofrequency) to dry fresh jujube and its polysaccharides (PS) and the effects of drying methods on the physical-, chemical-, structural- and anti-oxidant properties of PS were studied. It was concluded that radio frequency drying is a potential polysaccharide extraction and drying technique which can provide a theoretical basis for its industrial production.
I have several minor but important suggestions to be done for the improvement of the MS leading to its possible publication in the said journal:
1)Explain the terminology "qi" for the general readers in simple language.
2)Specify the model and the producer of radiofrequency equipment
3)Specify the model and the producer of HPLC
4)Specify the model and the producer of HPSEC
5)Specify the model and the producer of rheometer
6)In the statement at line numbers 239 and 240, please provide suitable reference in favour of your statement
7)In line numbers 298 and 299, the word "these" may be removed from the foot note of Figure 3.
8)"Conclusion" may be a better expressive term, than "Conclusions" for this particular MS, as there are not much conclusions out of the designed study.
Author Response
I have gone through the MS entitled "Comparative evaluation of physicochemical properties, micro-structure, and antioxidant activity of jujube polysaccharides subjected to hot air, infrared, radiofrequency and freeze drying". The authors used four drying methods (hot air, freezing, infrared, and radiofrequency) to dry fresh jujube and its polysaccharides (PS) and the effects of drying methods on the physical-, chemical-, structural- and anti-oxidant properties of PS were studied. It was concluded that radio frequency drying is a potential polysaccharide extraction and drying technique which can provide a theoretical basis for its industrial production.
I have several minor but important suggestions to be done for the improvement of the MS leading to its possible publication in the said journal:
Reviewer comment: 1. Explain the terminology "qi" for the general readers in simple language.
Our response: Thank you for your advice. We have explained the term.
Reviewer comment: 2. Specify the model and the producer of radiofrequency equipment
Our response: We have added equipment models and manufacturers in line 82-83.
Reviewer comment: 3. Specify the model and the producer of HPLC
Our response: We have added equipment models and manufacturers in line 109-111.
Reviewer comment: 4. Specify the model and the producer of HPSEC
Our response:. We have added equipment models and manufacturers in line 127-128.
Reviewer comment: 5. Specify the model and the producer of rheometer
Our response: We have added equipment models and manufacturers in 143.
Reviewer comment: 6. In the statement at line numbers 239 and 240, please provide suitable reference in favour of your statement
Our response: We have added a reference in 413-414.
Reviewer comment: 7. In line numbers 298 and 299, the word "these" may be removed from the foot note of Figure 3.
Our response: We have removed the word.
Reviewer comment: 8."Conclusion" may be a better expressive term, than "Conclusions" for this particular MS, as there are not much conclusions out of the designed study.
Our response: Thank you very much for your kind reminder. We have revised the text.